

# NMR structure of the C-terminal domain of TonB protein from *Pseudomonas aeruginosa*

Jesper S. Oeemig[1,2,*], O.H. Samuli Ollila[1,3,*] and Hideo Iwaï[1]

[1] Research Program in Structural Biology and Biophysics, Institute of Biotechnology, University of Helsinki, Helsinki, Finland
[2] VIB Center for Structural Biology, Vlaams Instituut voor Biotechnologie (VIB), Vrije Universiteit Brussel, Brussels, Belgium
[3] Institute of Organic Chemistry and Biochemistry, Academy of Sciences of the Czech Republic, Prague, Czech Republic
* These authors contributed equally to this work.

## ABSTRACT

The TonB protein plays an essential role in the energy transduction system to drive active transport across the outer membrane (OM) using the proton-motive force of the cytoplasmic membrane of Gram-negative bacteria. The C-terminal domain (CTD) of TonB protein is known to interact with the conserved TonB box motif of TonB-dependent OM transporters, which likely induces structural changes in the OM transporters. Several distinct conformations of differently dissected CTDs of *Escherichia coli* TonB have been previously reported. Here we determined the solution NMR structure of a 96-residue fragment of *Pseudomonas aeruginosa* TonB (*Pa*TonB-96). The structure shows a monomeric structure with the flexible C-terminal region (residues 338–342), different from the NMR structure of *E. coli* TonB (*Ec*TonB-137). The extended and flexible C-terminal residues are confirmed by $^{15}N$ relaxation analysis and molecular dynamics simulation. We created models for the *Pa*TonB-96/TonB box interaction and propose that the internal fluctuations of *Pa*TonB-96 makes it more accessible for the interactions with the TonB box and possibly plays a role in disrupting the plug domain of the TonB-dependent OM transporters.

## INTRODUCTION

The periplasmic space resides between the outer membrane (OM) and cytoplasmic membrane (CM) of Gram-negative bacteria. The OM protects Gram-negative bacteria from environmental hazards such as antibiotics and detergents. At the same time, Gram-negative bacteria require rare essential nutrients such as iron, vitamins that are present in the extracellular environment at very low concentrations. Gram-negative bacteria have evolved active acquisition systems to pass the essential nutrients through OM and CM. Since there is no electrochemical gradient to power the active transport at the OM and no ATP in the periplasmic space, these transporters must extract energy from

Corresponding author
Hideo Iwaï, hideo.iwai@helsinki.fi

the CM. These transporters in the OM are termed TonB-dependent transporters (TBDT) because they presumably extract the energy from the proton-motive force (pmf) of the CM via the *trans*-periplasmic protein, TonB protein (Fig. 1A). The energy transduction is assumed to take place via the TonB complex anchored in CM and consisting of TonB, ExbB, and ExbD proteins. ExbB and ExbD are accessory proteins anchored in the CM that convey the pmf across the CM to TonB (*Celia et al., 2016*; *Clarke, Tari & Vogel, 2001*; *Krewulak & Vogel, 2008*; *Postle & Larsen, 2007*).

TonB protein mediates the energy transduction from the CM to TBDTs. TonB has an N-terminal transmembrane (TM) domain anchored in the CM (Fig. 1). The TM domain is followed by the central region mostly consisting of Pro-Glu and Pro-Lys repeats (Fig. 1B). Extended conformation of the central region allows the protein to span the periplasmic space between OM and CM (*Domingo Köhler et al., 2010*). The C-terminal domain (CTD) of TonB protein has a globular structure and interacts with a conserved TonB box motif located at the N-terminus of TBDTs (*Cadieux & Kadner, 1999*; *Pawelek et al., 2006*; *Peacock et al., 2005*; *Shultis et al., 2006*). Despite a wide variety of models for the energy transduction mechanism by TonB, it is still unclear how TonB protein works in the energy transduction that presumably causes the structural changes of the plug domain in TBDTs to facilitate the transport (*Celia et al., 2016*; *Chimento, Kadner & Wiener, 2005*; *Gresock et al., 2011*; *Klebba, 2016*; *Letain & Postle, 1997*; *Sverzhinsky et al., 2015*; *White et al., 2017*). The crystal structures of the complexes with two different outer membrane receptors from *Escherichia coli* BtuB and FhuA suggests direct interactions between β-sheets in the CTD of TonB protein and the TonB box (*Shultis et al., 2006*; *Pawelek et al., 2006*). Currently proposed models to induce the structural changes of the plug domain include the mechanical pulling of the plug domain via the interaction with TonB box and CTDs of TonB, which are supported by the atomic force microscopy and molecular dynamics (MD) simulation studies suggesting that the interaction is strong enough to remain stable during the mechanical unfolding of the plug domain (*Chimento, Kadner & Wiener, 2005*; *Gumbart, Wiener & Tajkhorshid, 2007*; *Hickman et al., 2017*). Furthermore, the interaction between a highly conserved positive charge at Arg166 of *Ec*TonB and the negative charge of Glu56 of FhuA receptor is proposed to be involved in the disruption of the plug domain (*Pawelek et al., 2006*).

Interestingly, several groups have reported distinct conformations of the differently dissected CTD of TonB protein from *E. coli* solved by X-ray crystallography or NMR spectroscopy (Fig. 1C). The crystal structures of *E. coli* TonB-CTD consisting of the last 85 or 77 residues were composed of an intertwined dimer conformation with three β-strands and one α-helix (PDB codes: 1IHR and 1QXX, respectively) (*Chang et al., 2001*; *Koedding et al., 2004*). Whereas a longer construct consisting of the last 92 residues (*Ec*TonB-92) was monomeric in solution, the crystal structure revealed dimerization via β-strands in the C-terminus (PDB: 1U07) (*Ködding et al., 2005*). Furthermore, TonB protein from *E. coli* (*Ec*TonB-137, PDB code: 1XX3) was monomeric as observed in solution NMR experiments (*Peacock et al., 2005*). CTDs of *Ec*TonB bound to outer membrane receptors BtuB and FhuA take a monomeric conformation (*Shultis et al., 2006*; *Pawelek et al., 2006*). Therefore, the biological relevance of the intertwined dimers
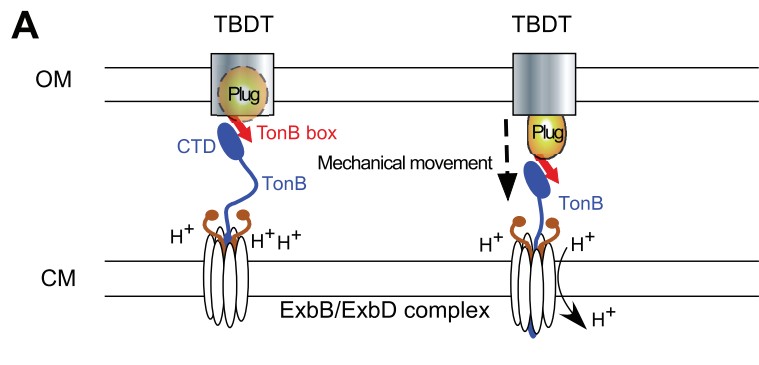

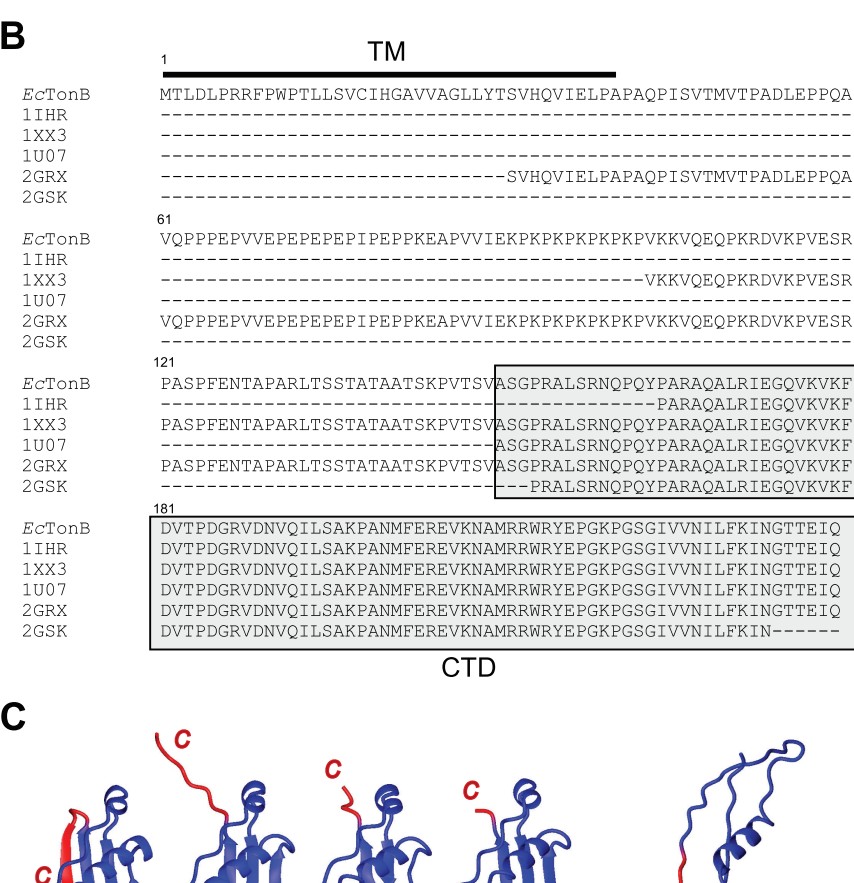

**Figure 1  TonB-dependent energy transduction system and structures of differently dissected _E. coli_ TonB.** (A) A schematic model of the TonB-dependent energy transduction system, which transduces the proton motive force of the CM to TonB-dependent transporters (TBDT) via TonB. (B) Sequence alignment presenting the full-length _E. coli_ TonB sequence and differently dissected CTDs used for the structure determination (labeled with PDB IDs). CTDs are highlighted in gray, and the TM region is indicated by a bar. (C) Ribbon drawings of previously reported _E. coli_ TonB CTD structures. The last β-strands at the C-terminal region with the largest conformational differences among the reported structures are colored in red. 1IHR forms intertwined dimers (_Chang et al., 2001_) and 1U07 forms dimers connected via the C-terminal extended strand (_Ködding et al., 2005_) in crystals.

found in the crystal structures of the shorter fragments of CTDs bearing the last 85 or 77 residues has not been clear (*Ködding et al., 2005*; *Koedding et al., 2004*; *Postle et al., 2010*). In contrast, Electron Spin Resonance (EPR) experiments suggested that a significant population of TonB exists as a dimer in solution while the monomeric form is bound to a TBDT (*Freed et al., 2013*). The major differences among the monomeric structures of *Ec*TonB are observed around the residues 235–239 (highlighted in red) at the C-terminal region (Fig. 1C). Whereas these residues in the NMR structure of *Ec*TonB-137 fold to an additional β-strand (β6) and forming an anti-parallel sheet with the preceding β5-strand, the same C-terminal end is either extended or largely invisible in the crystal structures (Fig. 1C). Intriguingly, β5-strand (residues 226–231) in *Ec*TonB-CTD directly interacts with TonB box of TBDTs in the crystal structures of *Ec*TonB-CTD/TBDT complexes (*Pawelek et al., 2006*; *Shultis et al., 2006*). The β5-strand is not accessible for the proposed interaction in the NMR structures of *Ec*TonB-137 (*Peacock et al., 2005*) due to the additional β6-strand. Therefore, β6-strand in *Ec*TonB-137 NMR structure has to be exchanged with the strand in TonB box when interacting with TBDTs (*Peacock et al., 2005*). In contrast, the recent solution NMR structure of the last 92 residues from *Helicobacter pyroli* TonB (*Hp*TonB-92) showed the disordered C-terminal region and the absence of β6-strand (*Ciragan et al., 2016*), which is more in line with the crystal structures of *Ec*TonB-CTD reported previously (*Ködding et al., 2005*; *Shultis et al., 2006*; *Pawelek et al., 2006*).

Here, we report the NMR structure and MD simulations of a CTD of TonB protein from *Pseudomonas aeruginosa* (*Pa*TonB-96) to investigate whether the plasticity of the C-terminal region observed for the reported structures of *Ec*TonB is a common feature of TonBs across different organisms.

# MATERIALS AND METHODS

## NMR sample preparation

The C-terminal 96 residues of a TonB protein from *P. aeruginosa* (UniProt: Q51368) was cloned from genomic DNA (ATCC-47085) as an N-terminal SUMO fusion protein, which was previously termed *Ps*TonB-96 (*Guerrero, Ciragan & Iwaï, 2015*). The construct expresses a His-tag SUMO fusion protein that produces a 99-residue protein after the SUMO tag removal, which we termed *Pa*TonB-96. *Pa*TonB-96 contains the last 96 residues of *Pa*TonB (247–342) and three residues (SHM) at the N-terminal end from the cloning site. The plasmid (pFGRSF15) coding the gene of *Pa*TonB-96 was transformed into *E. coli* ER2566 strain (New England Biolabs, Ipswich, MA, USA) for production of doubly $^{15}$N, $^{13}$C-labeled samples. The transformed *E. coli* cells were grown overnight at 30 °C in 50 mL LB medium supplemented with 25 μg/mL kanamycin. The cells were spun down at 900 × g for 15 min and gently re-suspended in two L pre-warmed M9 medium supplemented with 25 μg/mL kanamycin, containing $^{15}$NH$_4$Cl and $^{13}$C$_6$-D-glucose as sole nitrogen and carbon sources, respectively. The cells were grown at 37 °C until an OD$_{600}$ of 0.6. Then, the temperature was lowered to 30 °C for protein expression. The protein expression was induced with a final concentration of one mM isopropyl β-D-1-thiogalactopyranoside (IPTG). The cells were incubated for additional 5 h before

harvesting by centrifugation at 4,900 × g at 4 °C for 15 min. The labeled $Pa$TonB-96 was purified following the previously published protocol (*Guerrero, Ciragan & Iwaï, 2015*). The purified protein was dialyzed against 20 mM sodium phosphate buffer (pH 6.0). The protein solution of $Pa$TonB-96 was concentrated to one mM for NMR analysis using a centrifugal device. The final sample volume was 250 μL containing 10% $D_2O$.

## NMR measurements

NMR measurements for the structure determination were recorded on Varian INOVA 800 MHz equipped with a cryogenically cooled five mm probe head. For the sequential backbone assignment, a standard set of double and triple resonance NMR spectra were recorded at 25 °C, including [$^1$H, $^{15}$N]-HSQC, HNCO, HNCA, HNCACB, HN(CO)CA, HN(CA)CO, and CBCA(CO)NH (*Sattler, Schleucher & Griesinger, 1999*). The aliphatic side-chain assignment was carried out using [$^1$H, $^{13}$C]-HSQC, HCCH-COSY, ct-[$^1$H, $^{13}$C]-HSQC, HBHA(CO)NH, H(CCCO)NH, (H)CC(CO)NH, $^{15}$N-resolved [$^1$H, $^1$H]-TOCSY, and $^{15}$N-edited [$^1$H-$^1$H]-NOESY spectra. The assignments for the aromatic side-chains were based on the spectra of (HB)CB(CGCD)HD, (HB)CB (CGCDCE)HE, aromatic region ct-[$^1$H, $^{13}$C]-HSQC and $^{13}$C-edited [$^1$H,$^1$H]-NOESY. Data was acquired using VnmrJ (Varian Inc., Palo Alto, CA, USA) and data were processed using the NMRpipe software (*Delaglio et al., 1995*).

NMR measurements for the backbone dynamics were recorded on a Bruker Avance 850 MHz equipped with a cryogenically cooled probe head. The longitudinal ($T_1$) and transverse ($T_2$) relaxation rates and $^{15}$N{$^1$H}-heteronuclear nuclear Overhauser effects (NOEs) for backbone $^{15}$N atoms were determined at 25 °C using the well-established NMR experiments (*Barbato et al., 1992*; *Kay, Torchia & Bax, 1989*). $T_1$($^{15}$N) and $T_2$($^{15}$N) relaxation times were determined using the following delay times: 10, 50, 100, 200, 300, 500, 800, 1,000, 1,200, and 2,000 ms for $T_1$ and 16, 64, 96, 128, 156, 196, 224, and 256 ms for CMPG pulse train with one ms interval for $T_2$ relaxation rates, respectively. Relaxation times were obtained by fitting a single exponential decay to peak intensity values: $I(t) = I_0 \times \exp(-t/T_1)$ or $I_0 \times \exp(-t/T_2)$, where $I(t)$ is the peak volume at a time $t$. Heteronuclear $^{15}$N{$^1$H}-NOEs were obtained with a relaxation delay of 5 s with or without saturation of protons. Heteronuclear $^{15}$N{$^1$H}-NOEs (η) were determined from the volumes of the HSQC signals using the ratio of $\eta = I/I_0$. The relaxation data were processed and analyzed using Bruker Dynamic Center (Version 2.1.8; Bruker Inc., Billerica, MA, USA).

## Solution NMR structure determination

The sequence-specific resonance assignment was performed with standard methods using triple resonance NMR experiments and CcpNmr Analysis software (version 2.4.1) (*Vranken et al., 2005*). The chemical shift values from the sequential resonance assignment were used together with NOE peak lists for the structure calculation with the program CYANA 3.0 (*Mumenthaler et al., 1997*; *Güntert & Buchner, 2015*). NOE distance restraints were obtained from 3D $^{15}$N- and $^{13}$C-edited [$^1$H, $^1$H]-NOESY spectra with 80-ms mixing

time. The conformations of prolines were checked based on CB-CG chemical shift before the structure calculation (*Schubert et al., 2002*). All prolines were predicted to be in *trans* conformation except P299 that was set to *cis*-conformation. The three-dimensional NMR conformers were generated using CYANA 3.0, based on the automated NOESY cross peaks assignment (*Güntert, 2004*; *Güntert, Mumenthaler & Wüthrich, 1997*). The restrained energy minimization of the final 20 best conformers was performed using AMBER 14 (*Pearlman et al., 1995*), and the structures were validated with PSVS 1.5 (*Bhattacharya, Tejero & Montelione, 2007*). The structural statistics are summarized in Table 1.

## MD simulation

The MD simulations were performed with Gromacs5 (*Abraham et al., 2015*) by using Amber ff99SB-ILDN force field and tip4p water model (*Jorgensen et al., 1983*; *Lindorff-Larsen et al., 2010*). The first 10 ns from total 400-ns simulation was considered to be the equilibrium period by monitoring the protein root-mean-square-deviation, inertia tensor eigenvalues, and rotation angles. The rest was used for the analysis since the first 10 ns of simulation trajectories was sufficiently enough to remove the significant fluctuations in these parameters (*Ollila, Heikkinen & Iwaï, 2018*). The NMR structure determined in this work was used as the initial structure and the secondary structures remained unchanged during the entire simulation period. The temperature was coupled to 25 °C with v-rescale thermostat (*Bussi, Donadio & Parrinello, 2007*), and the pressure was isotropically set to one bar using Parrinello-Rahman barostat (*Parrinello & Rahman, 1981*). The time-step was two fs, Lennard-Jones interactions were cut-off at one nm, PME (*Darden, York & Pedersen, 1993*; *Essmann et al., 1995*) was used for electrostatics, and LINCS (*Hess, 2008*) was used to constrain all bond lengths. The simulation data is available from the Zenodo repository (*Ollila, 2018a*).

The analysis and interpretation of $^{15}$N spin relaxation times are described in detail elsewhere (*Ollila, Heikkinen & Iwaï, 2018*). Briefly, rotational correlation functions of the backbone N–H bonds were calculated from the simulation data and the spin relaxation times were calculated using the Redfield equations (*Abragam, 1961*; *Kay, Torchia & Bax, 1989*; *Ollila, Heikkinen & Iwaï, 2018*). Before the spin relaxation time calculation, the overestimated overall rotational diffusion in the MD simulation due to the water model was corrected by dividing the rotational diffusion coefficients around all inertia axes with a factor of 1.2, assuming that the protein rotates as an anisotropic rigid body. This scaling factor was found to be capable of reproducing the $^{15}$N spin relaxation times in good agreement with the experimental data of *Pa*TonB-96 for tip4p water model (*Ollila, Heikkinen & Iwaï, 2018*). The computer codes used for the analysis and the related data are available (*Ollila, 2018b*, *2018c*).

## RESULTS

### NMR solution structure of *Pa*TonB-96

The protein consisting of the C-terminal 96 residues of TonB from *P. aeruginosa* (*Pa*TonB-96) was previously identified as a minimal domain that was soluble when

**Table 1** Structural statistics of the energy-minimized NMR structure of *Pa*TonB-96.

| | *Pa*TonB-96[a] |
|---|---|
| **Completeness of resonance assignments (%)[b]** | |
| Backbone | 98.5 |
| Side chain, aliphatic | 97.0 |
| Side chain, aromatic | 88.0 |
| **Distance restraints** | |
| Total | 1,698 |
| Intraresidue ($i = j$) | 465 |
| Sequential ($|i-j| = 1$) | 483 |
| Medium range ($1 < |i-j| < 5$) | 198 |
| Long range ($|i-j| \geq 5$) | 552 |
| No. of restraints per residue | 17.2 |
| No. of long-range restraints per residue | 5.6 |
| **Residual restraint violations** | |
| Average no. of distance violation per structure | |
| 0.1–0.2 Å | 1 |
| >0.2 Å | 0 (max. 0.14) |
| Average no. of dihedral angle violations per structure | |
| >2.5° | 0 |
| **Model quality[c]** | |
| Rmsd backbone atoms (Å) | 1.0 |
| Rmsd heavy atoms (Å) | 1.6 |
| Rmsd bond lengths (Å) | 0.014 |
| Rmsd bond angles (°) | 2.1 |
| **MolProbity Ramachandran statistics[c]** | |
| Most favored regions (%) | 97.4 |
| Allowed regions (%) | 2.5 |
| Disallowed regions (%) | 0.1 |
| **Global quality scores (raw/Z score)[c]** | |
| Verify3D | 0.34/−1.93 |
| ProsaII | 0.41/−0.99 |
| PROCHECK (ϕ–ψ) | −0.27/−0.75 |
| PROCHECK (all) | −0.23/−1.36 |
| MolProbity clash score | 0.18/1.49 |
| **Model contents** | |
| Ordered residue ranges | 247–258, 261–320, 323–338 |
| Total no. of residues | 99 |
| BMRB accession number | 34235 |
| PDB ID code | 6FIP |

**Notes:**
[a] Structural statistics computed for the ensemble of the 20 deposited NMR conformers.
[b] Calculated from the expected number of resonances, excluding highly exchangeable protons (N-terminal, Lys, amino and Arg guanidino groups, hydroxyls of Ser, Thr, and Tyr), carboxyl groups of Asp and Glu, and nonprotonated aromatic carbons. Backbone atoms: HN, N, Cα, Cβ, Hα, and C.'
[c] Calculated using PSVS version 1.5 (*Bhattacharya, Tejero & Montelione, 2007*).

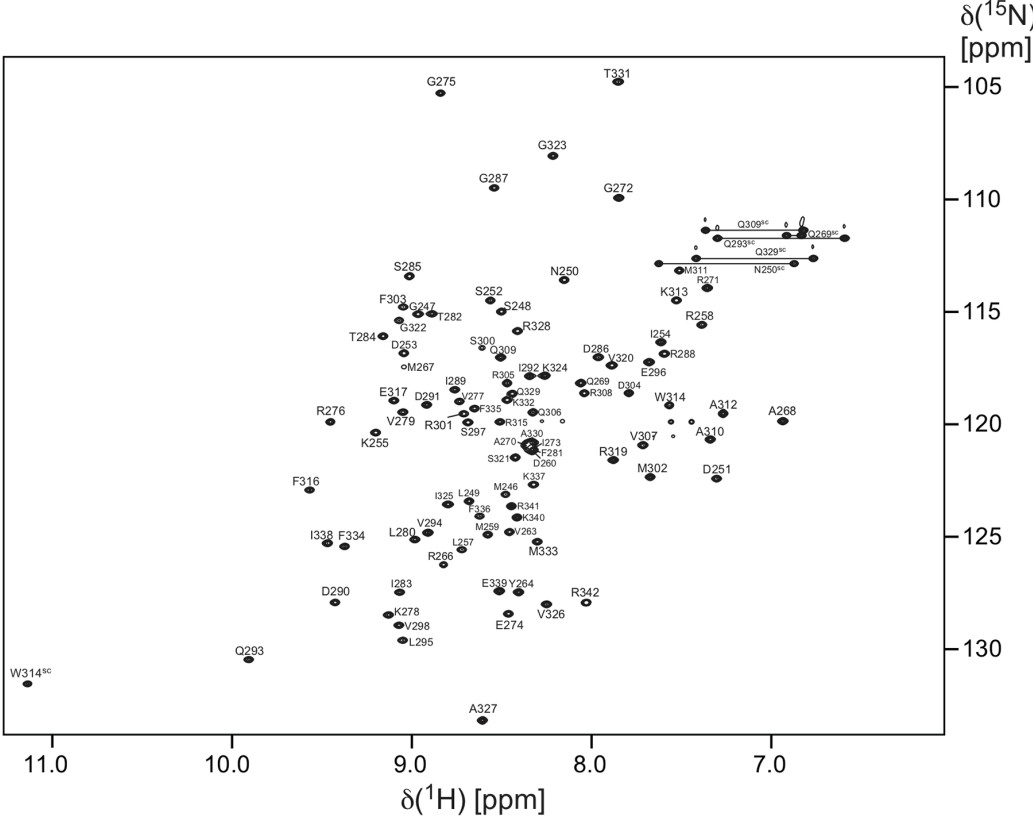

**Figure 2 Two-dimensional [¹H, ¹⁵N]-HSQC spectrum of one mM _Pa_TonB-96.** The spectrum was recorded at the ¹H frequency of 800 MHz at 25 °C. The residue number and single letter code for amino acid types indicate the assignments. Trp HNᵉ, Gln and Asn side chain amide resonances are marked by "sc."

expressed using an _E. coli_ expression system, while other shorter fragments were not soluble even with several different fusion partners (_Guerrero, Ciragan & Iwaï, 2015_). This fragment has 45% identity (40/88 residues) to _Ec_TonB-92, suggesting a similar structure to _Ec_TonB. [¹H, ¹⁵N]-HSQC spectrum of _Pa_TonB-96 shows the well-dispersed NMR signals for amide groups (Fig. 2), indicating a globular folded conformation. All of the main-chain chemical shifts were assigned except for S244 and Hᴺ of H245. 95.2% of the expected side-chains were assigned. The assigned chemical shifts were used for the automatic analysis of NOE peaks with the CYANA to calculate the NMR structure. Figure 3 shows the lowest energy solution NMR structure of _Pa_TonB-96 with the secondary structure elements and a superposition of the 20 lowest energy conformers. The structural statistics of the 20 NMR conformers are summarized in Table 1. _Pa_TonB-96 adopts a mixed α/β structure with the topology of βαββαββ. The structure consists of a β-sheet composed of three anti-parallel β-strands (β2, β3, and β5), a β-sheet composed of two short β-strands (β1 and β4) and two short α-helices (αI and αII). The structural coordinates and the chemical shifts have been deposited in the Protein Data Bank (http://www.rcsb.org/, PDB code: 6FIP) and BMRB (http://www.bmrb.wisc.edu/, accession number: 34235).

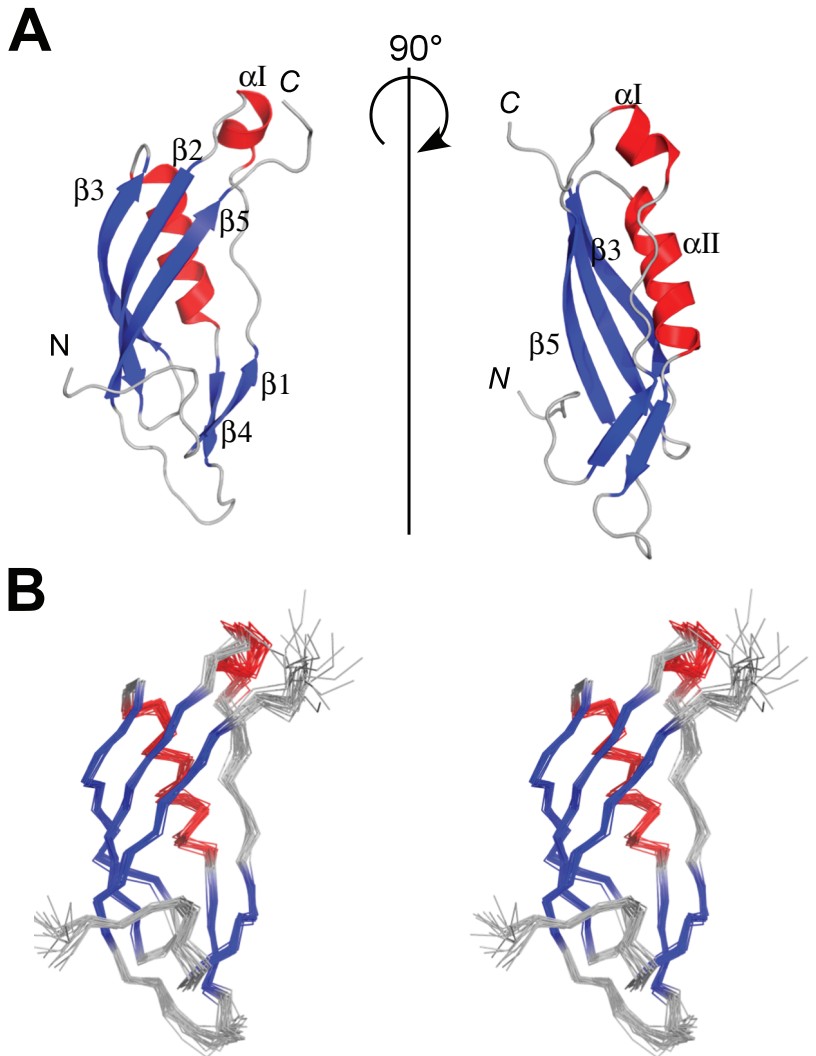

**Figure 3 NMR structures of *Pa*TonB-96.** (A) Ribbon drawings of the lowest energy conformer of the *Pa*TonB-96 structures showing the secondary structure elements. (B) Stereoview of an ensemble of the 20 lowest energy NMR conformers. Red and blue color are used for α-helices and β-sheets, respectively. N and C indicate N- or C-termini, respectively. Figures are generated with PyMol (*Schrodinger, 2015*).

## Comparison of the NMR structures between *E. coli* and *P. aeruginosa*

The primary and secondary structures of *Pa*TonB-96 were compared with the structure of *Ec*TonB-137 (PDB: 1XX3) (*Peacock et al., 2005*) (Fig. 4). The length of *Pa*TonB-96 is similar to that of the globular part of *Ec*TonB-137 (88 residues). Notable structural differences are observed around the C-terminal end. The C-terminal end of *Ec*TonB-137 forms an anti-parallel β-strand (β6) with β5-strand to constitute a β sheet. In contrast, the C-terminal end in *Pa*TonB-96 is unstructured and exhibits an extended flexible conformation as also seen from the spin relaxation data (Fig. 5). The difference can be easily explained by the shorter C-terminal end after the β5-sheet in *Pa*TonB-96, being too short to form an additional β-strand. The shorter C-terminal end in *Pa*TonB-96 is compensated by the longer loop between β4 and β5-strands with additional five residues,

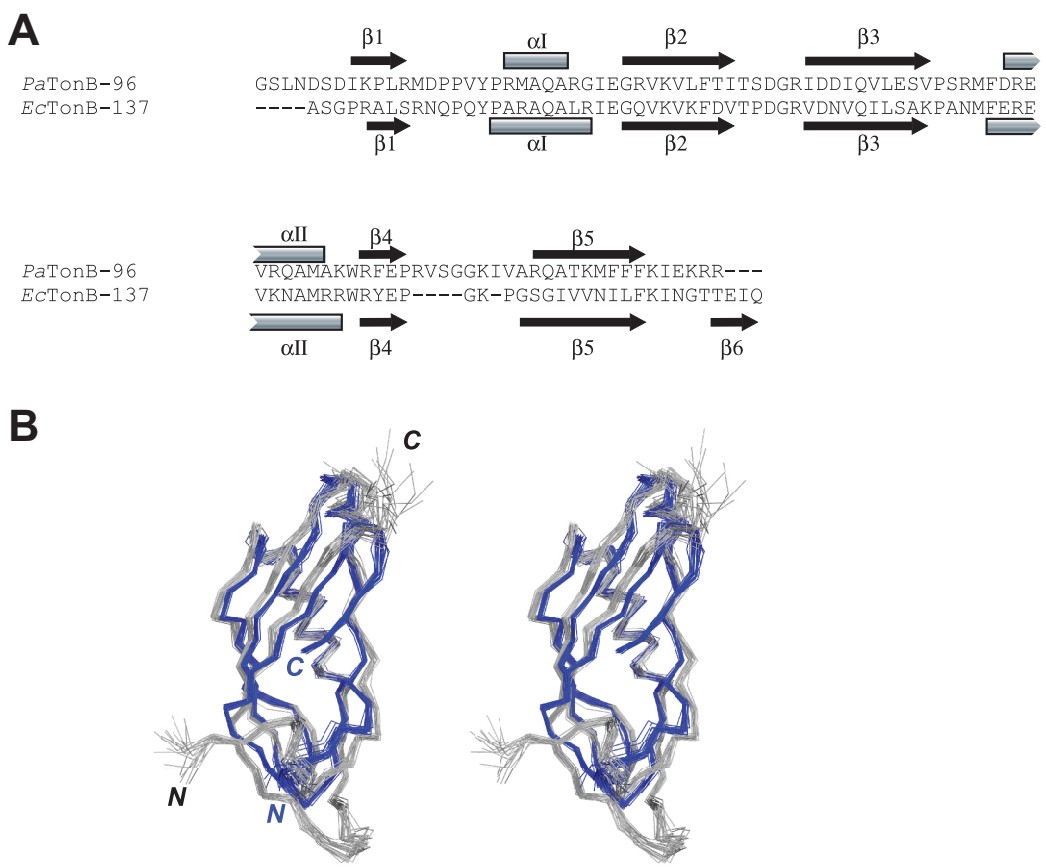

**A**

**αII** **β4** **β5**

*Pa*TonB-96
*Ec*TonB-137

**Figure 4 Comparison between *Pa*TonB-96 and *Ec*TonB-137.** (A) The sequence comparison between *Pa*TonB-96 and the last 90 residues of *Ec*TonB-137 with the secondary structure elements. (B) Stereoview of the superposition of the 20 NMR structures of *Pa*TonB-96 (gray) and residues 151–239 of the 20 NMR structures of *Ec*TonB-137 (PDB: 1XX3, blue). N and C stand for the N- and C-termini, respectively.

compared with the structured 88-residue region of *Ec*TonB-137. Higher mobility of this longer loop is also confirmed by the $^{15}N$ relaxation measurement and MD simulation (Fig. 5). Interestingly, the extended and disordered conformation of the C-terminal end in *Pa*TonB-96 more closely resembles the crystal structures of *Ec*TonB-92 (PDB: 1U07) or the structures from TonB/TBDT complexes (PDB: 2GRX and 2GSK) (Fig. 1).
The interaction between β5-strand of TonB and the TonB box in TBDTs has been proposed to be essential for the TonB-mediated energy transfer (*Pawelek et al., 2006*; *Shultis et al., 2006*). Such interaction would be hindered by the additional β6-stand found in *Ec*TonB-137. Thus, our results suggest that β5-strand is more accessible for the proposed interactions with TonB box in *Pa*TonB-96 where the β6-strand is absent.

## Structural motions in *Pa*TonB-96

Fast dynamics (picosecond to nanosecond) of proteins has been commonly investigated by measuring the spin relaxation times ($T_1$, $T_2$, and heteronuclear NOEs) of backbone $^{15}N$ atoms (*Jarymowycz & Stone, 2006*; *Van Den Bedem & Fraser, 2015*). Here, we characterize the structural dynamics of *Pa*TonB-96 by combining $T_1$, $T_2$, and

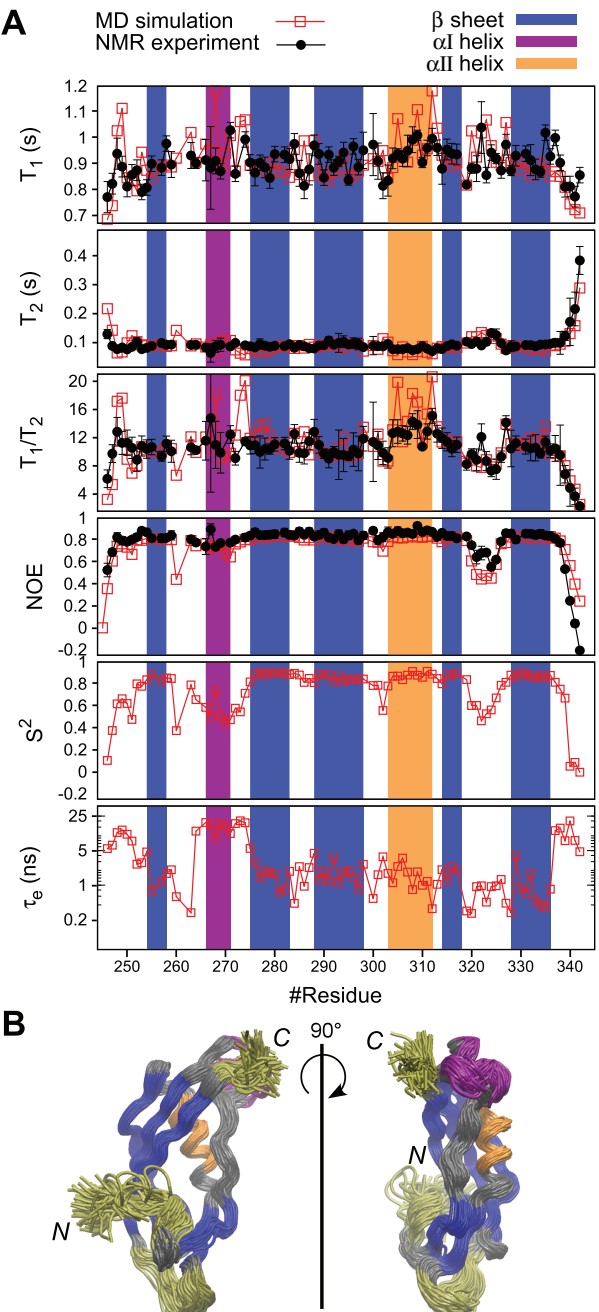

**Figure 5 MD simulation of *Pa*TonB-96 and comparison with the experimental data.** (A) The comparison of the experimental and simulated $^{15}$N relaxation parameters and the internal motions presented by $S^2$ and $t_e$ obtained from the MD simulation. The regions for β-sheets, αI, and αII are highlighted in blue, purple, and orange, respectively. (B) A superposition of snapshots of the structure from the MD simulation trajectory indicating conformational fluctuations during the MD simulation. The residues with enhanced flexibilities are colored in yellow. The aI-helix with orientation fluctuations is colored in purple. Rotational diffusion coefficient values obtained from the MD simulation of the experimental spin relaxation data are, $D_{xx} = 1.51 \pm 0.01$, $D_{yy} = 1.72 \pm 0.03$, and $D_{yy} = 3.79 \pm 0.03$ (rad$^2 \cdot 10^7$/s). These result in $D_{av} = (D_{xx} + D_{yy} + D_{zz})/3 = 2.3 \pm 0.02$ (rad$^2 \cdot 10^7$/s), $\tau_c = (6D_{av})^{-1} = 7.2 \pm 0.02$ (ns), $A = 2D_{zz} - (D_{yy} + D_{xx}) = 4.4$ and $R = D_{yy} - D_{xx} = 0.2$, where $A$ is the axiality and $R$ is the rhombicity. The overlay of the trajectories was produced by vmd (*Humphrey, Dalke & Schulten, 1996*).

heteronuclear NOEs for backbone $^{15}$N atoms and MD simulations (*Ollila, Heikkinen & Iwaï, 2018*). The measured spin relaxation times for *Pa*TonB-96 are shown together with the MD simulation results (Fig. 5A). The good agreement with the experimental data allows us to use the MD simulation trajectory to interpret the timescales for the overall rotational diffusion ($\tau_c$), effective correlation times for internal mobility ($\tau_{eff}$), and order parameters ($S^2$). The overall rotational diffusion coefficients around inertia axes (see the caption of Fig. 5) show that *Pa*TonB-96 has high axiality ($A = 4.4$), but with a rather low rhombicity ($R = 0.2$) suggesting that the protein has an approximately ellipsoidal shape. The MD simulation analysis estimates $\tau_c = 7.2$ ns for the timescale of average overall rotational diffusion. This is in line with 6.9 ns estimated from the $T_1/T_2$ values using the model-free approach (*Kay, Torchia & Bax, 1989*; *Ollila, Heikkinen & Iwaï, 2018*) as well as with the values in the literature for monomeric proteins with the similar molecular weight (*Krishnan & Cosman, 1998*). We thus conclude that *Pa*TonB-96 is monomeric in solution as was the previously published NMR structure for *Ec*TonB-137 (*Peacock et al., 2005*).

The results reveal that there are four regions with enhanced internal motions in *Pa*TonB-96 (Fig. 5). The regions with notable conformational fluctuations are colored in yellow and purple in the overlaid snapshots from the MD simulation (Fig. 5). The first few N-terminal residues exhibit low order parameters and long effective correlations times related to the enhanced conformational fluctuations, suggesting that the N-terminal region is already the beginning of the flexible central region of the TonB protein that connects between TM region and CTD. The MD simulation revealed orientational fluctuations in αI helix (purple in Fig. 5B). This fluctuation also influenced the order parameters and effective correlation times of the neighboring residues in the MD simulation (Fig. 5A). However, the changes were not observed in the $^{15}$N spin relaxation analysis, presumably because the $^{15}$N relaxation is not sensitive to the time scale of the fluctuation. Residues 320–326 in the loop between β4 and β5-strands indicate some enhanced conformational fluctuations, which were also detectable by the $^{15}$N spin relaxation analysis and order parameters but does not affect the effective correlation times (Fig. 5A). The last five residues of the C-terminal end (residues 338–342) showed the enhanced conformational fluctuations, characterized with lower order parameters, long effective correlation times, and changes in the $^{15}$N spin relaxation rates, which are consistent with the experimental $^{15}$N spin relaxation data. The flexible residues at the C-terminal end are in close contact with the αI helix, indicating orientational fluctuations as described above.

## Modelling of *Pa*TonB and TonB box interactions

It is believed that TonB conveys the CM chemical potential to TBDTs by structural changes via the interaction between CTD of TonB and TonB box, a short stretch of peptide chain in the N-terminus of plug domain of TBDTs (*Cadieux & Kadner, 1999*; *Pawelek et al., 2006*; *Peacock et al., 2005*; *Shultis et al., 2006*) (Fig. 1A). In the crystal structure of *Ec*TonB-93/BtuB complex, *Ec*TonB orients the αI helix against the plug domain and forms a parallel β-sheet interaction with the TonB box of BtuB (*Shultis et al., 2006*) (Fig. 6A). We created a hypothetical model of a complex between *Pa*TonB-96 and TBDT by

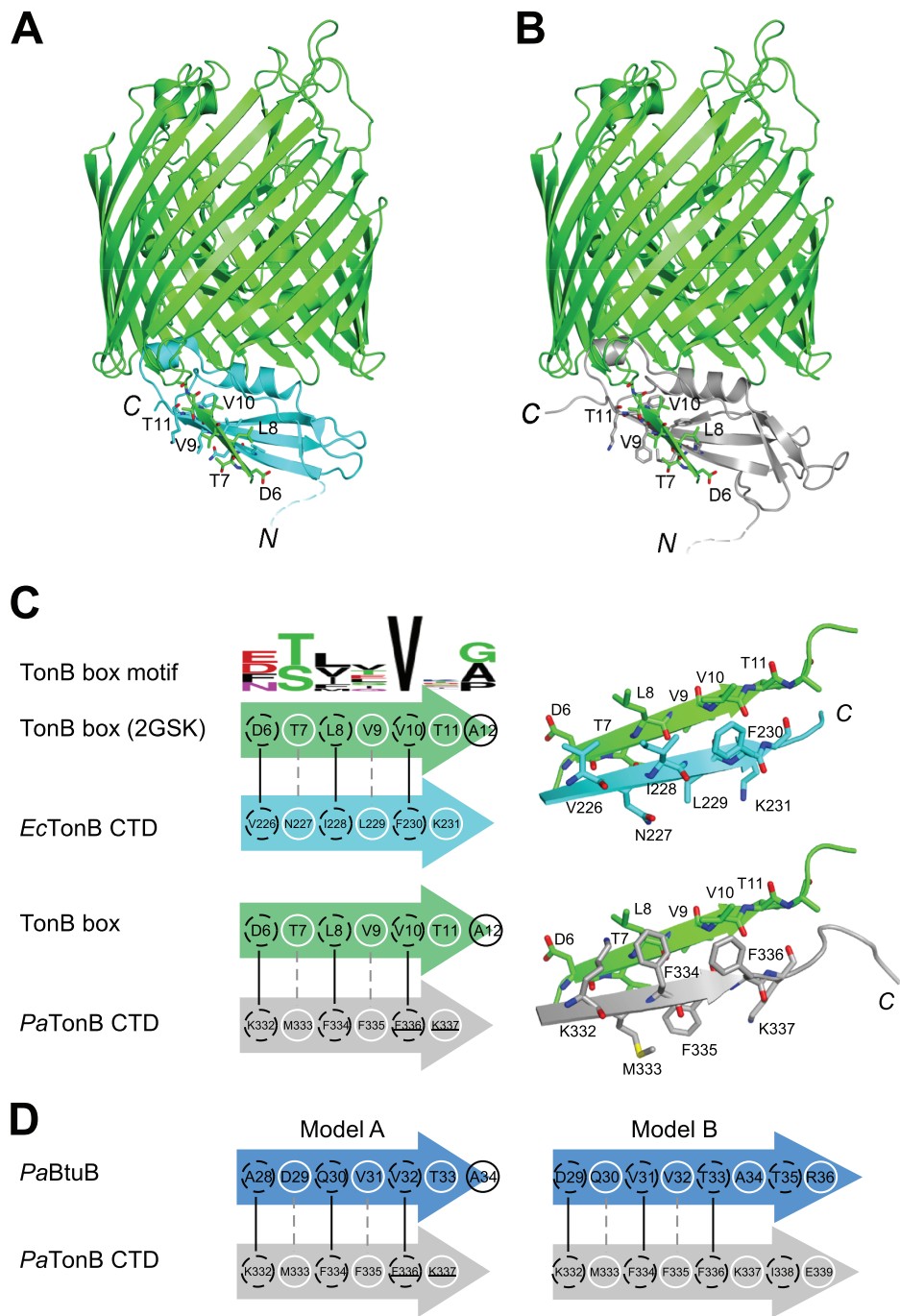

Figure 6 **Models for interactions between TonB and TBDTs.** (A) The crystal structure of *Ec*TonB-93 (cyan) bound to BtuB transporter (green, PDB: 2GSK). (B) A cartoon model of the complex of *E. coli* BtuB transporter and *Pa*TonB-96 (gray) created by the superposition of CTDs of the TonB proteins. (C) Illustrations depicting the interactions between β5-strand of *Ec*TonB-93 (cyan) or *Pa*TonB-96 (gray) with the TonB box of BtuB transporter from *E. coli* (green). The consensus TonB box sequences based on 57 sequences from UniProtKB/Swiss-Prot (PROSITE: PS00430) is shown at the top. (D) Two hypothetical interaction models for *Pa*TonB/TonB box from probable *P. aeruginosa* BtuB (Uniprot: Q9I473). White circles and dotted black circles indicate exposed and buried sides of the model, respectively.



superimposing *Pa*TonB-96 to residues 153–233 of *Ec*TonB in the crystal structure of the complex because no structure of TBDTs from *P. aeruginosa* is available (Fig. 6B). β5-strand (residues 226–231) of *Ec*TonB-93 and TonB box of BtuB forms the parallel β-strands, of which interactions are mostly hydrophobic (Fig. 6C). The modeled structure of *Pa*TonB-96/BtuB complex indeed resembles the conserved TonB box interactions observed with the *Ec*TonB/BtuB complex, supporting that the modeled structure is plausible. The highly conserved Val10 in the TonB box is located next to phenylalanine in both TonB structures (Phe230 and Phe336 in *Ec*TonB-93 and *Pa*TonB-96, respectively), suggesting that this hydrophobic interaction might be critical for the TonB/TBDT complexes (Fig. 6C).

## DISCUSSION

We are interested whether the structural features found in TonB and TonB/TBDTs complexes from *E. coli* are shared among other Gram-negative organisms. In the modeled structure of *Pa*TonB-96/*Ec*BtuB complex, the positive charge at the beginning of β5-strand (Lys332 in *Pa*TonB-96) could complement the negative charge of Asp6 in the TonB box, of which negative charge can be found in the half of the TonB box consensus motif from various Gram-negative organisms (Fig. 6C). This might suggest that the positive-negative charge interaction could play an important role in the specific recognition of TonB box. However, putative BtuB sequences from *P. aeruginosa* do not contain the exact consensus TonB box motif shown in Fig. 6C. Based on the observation from the model of *Pa*TonB-96/*Ec*BtuB complex, we searched a potential TonB box sequence in a putative BtuB sequence (*Pa*BtuB) (Uniprot: Q9I473). We found a stretch of the sequence "DQVVTATR" (residues 29–36) at the N-terminal region of the possible plug domain region, with which *Pa*TonB might interact, and hypothesized two interaction models (Fig. 6D). In the model A, we assumed Val32 in the *Pa*BtuB as the highly conserved Val in the TonB box motif. In this model, the negative charge of Asp29 could be located in the vicinity of Lys332 and the sequence of "VVTA" is identical to the *Pa*TonB-96/*Ec*BtuB model. In the other model (model B), we considered Asp29 as the key residue in the alignment. The highly conserved Val in the TonB box is now replaced by Ala in this model, but the negative charge at Glu339 in *Pa*TonB could be better compensated by Arg36 in *Pa*BtuB if the extended the last β-sheet of *Pa*TonB.

It is noteworthy that similar interactions are not plausible for the solution NMR structure of *Ec*TonB-137 due to the steric hindrance by the presence of an additional β6-strand. The β6-strand needs to be disrupted and exchanged to the TonB box for the similar interaction as observed in the crystal structures of the two TonB/TBDT complexes (*Pawelek et al., 2006*; *Shultis et al., 2006*). The solution NMR structure of *Pa*TonB-96 elucidated in this work is accessible to the TonB box without the strand exchange, because the disordered and extended C-terminal end does not interfere with the interaction site. This open conformation at the C-terminus is also observed in the NMR structures of *Hp*TonB-92 (*Ciragan et al., 2016*) and TonB-like protein, HasB from *Serratia marcescens* (*Amorim et al., 2013*). Thus it might be a more common structure among CTDs of TonB proteins across different organisms.

Our hypothetical model of *Pa*TonB-96 interactions with the TonB box and crystal structures of TonB/TBDT complexes suggests that αI–helix positions toward the plug domain located in the TBDT barrel (Fig. 6A) (*Pawelek et al., 2006*; *Shultis et al., 2006*). It was previously proposed that the positively charged Arg166 residue in αI–helix of *Ec*TonB-CTD could interact with the negatively charged Glu56 in plug domain of FhuA receptor and disrupt its structure, thereby lowering the energy barrier required for the pore opening (*Pawelek et al., 2006*). Interestingly, the MD simulation detected orientational fluctuations of αI–helix in *Pa*TonB-96 (Fig. 5), which also contains positively charged Arg271. Such fluctuations might further facilitate the proposed disruption due to the interactions between Arg271 in *Pa*TonB and the negatively charged residues in the plug domain of TBDTs. The flexible loop between β4 and β5 strands found in *Pa*TonB-96 is located distantly from BtuB (Fig. 6B). Therefore, the modeled complex of *Pa*TonB/BtuB does not directly indicate any structural role for this flexible loop.

## CONCLUSIONS

We report the solution structure of *Pa*TonB-96, which is largely similar to the previously reported monomeric NMR structure of *Ec*TonB-137 (PDB code: 1XX3) (*Peacock et al., 2005*) except for the C-terminal end. Whereas the C-terminal region of NMR structures of *Ec*TonB-137 formed an additional anti-parallel β6-strand with β5-strand, the C-terminal end of *Pa*TonB-96 has extended flexible conformation, which resembles the crystal structures of TonB proteins interacting with TBDTs (*Pawelek et al., 2006*; *Shultis et al., 2006*). The absence of the last β6 strand in *Pa*TonB-96 structure suggests that the β5-strand is more accessible for the interaction with TonB box than in *Ec*TonB-137, for which the strand exchange is required. Furthermore, the structural model suggests that the electrostatic interactions between *Pa*TonB-96 and TonB box might be more favorable than for *Ec*TonB. Based on the structural model, we identified a potential TonB box sequence in *Pa*BtuB. We also speculate that the orientation fluctuations observed in the αI–helix of *Pa*TonB-96 detected in MD simulation could lower the energy barrier of the suggested channel opening by disrupting the plug domain structure within the TBDT barrel when the C-terminus of TonB is bound to TonB box. The NMR structures and *Pa*TonB-96 could thus guide further experimental analysis to unveil the structural basis of the mechanism of TonB-dependent energy transduction.

## ACKNOWLEDGEMENTS

We thank H.A. Heikkinen and S. Jääskeläinen for their technical assistance.

### Funding

This work was supported by Sigrid Jusélius Foundation, the Academy of Finland (131413, 137995 and 277335) and CSC-IT Center for Science Ltd. Finland (the allocation of computational resources). The Finnish Biological NMR center is supported by Biocenter Finland and HiLIFE-INFRA. The funders had no role in study design, data collection and analysis, decision to publish, or preparation of the manuscript.

## Grant Disclosures

The following grant information was disclosed by the authors:

Sigrid Jusélius Foundation, the Academy of Finland: 131413, 137995, and 277335.

CSC-IT Center for Science Ltd. Finland.

Biocenter Finland and HiLIFE-INFRA.

## Competing Interests

The authors declare that they have no competing interests.

## Author Contributions

- Jesper S. Oeemig performed the experiments, analyzed the data, prepared figures and/or tables, authored or reviewed drafts of the paper, approved the final draft.
- O.H. Samuli Ollila conceived and designed the experiments, performed the experiments, analyzed the data, prepared figures and/or tables, authored or reviewed drafts of the paper, approved the final draft.
- Hideo Iwaï conceived and designed the experiments, performed the experiments, analyzed the data, prepared figures and/or tables, authored or reviewed drafts of the paper, approved the final draft.

## Data Availability

Protein Data Bank: Dataset ID: D_1200008387 and PDB ID: 6FIP. This data can also be found in a Supplemental File.

BMRB (http://www.bmrb.wisc.edu/, accession number: 34235).

Ollila, O.H. Samuli. (2017). MD simulation data for *Pseudomonas aeruginosa* TonB-CTD, Amber ff99SB-ILDN, tip4p, 298K, Gromacs [Data set]. Zenodo. http://doi.org/10.5281/zenodo.1010416.

## Supplemental Information

Supplemental information for this article can be found online at http://dx.doi.org/10.7717/peerj.5412#supplemental-information.

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
