# Peer review of "NMR structure of the C-terminal domain of TonB protein from Pseudomonas aeruginosa"

_PeerJ, doi:10.7717/peerj.5412_

## Round 0.1 · original submission · Minor Revisions

The reviewers have clearly outlined several points that require revision and/or attention. Please respond clearly to each.

·

Basic reporting

In this paper, the authors outline the structure determination of Pseudomonas aeruginosa TonB C-terminal domain, confirmed it to be monomeric, and explored its dynamics at the atomic level.

The paper is generally very well written. There is a good introduction, with relevant literature referred to. However, a schematic figure would be beneficial here, illustrating the general set-up of TonB proteins, in the context of the OM and CM, and their relationship to TBDTs. This would be very useful for readers unfamiliar with this system.

The plug domain is first mentioned (lines 67-68), along with the ‘mechanical model’, but no context has been given for these terms. Could the authors introduce these ideas i.e. how the plug domain of the TBDT is proposed to mechanically unfold in the ‘mechanical model’?

The chemical shifts and structures have been deposited in public databases (lines 202-204), but have the restraints used in the calculations, NOESY spectrum peaks, and relaxation data been deposited also?

Line 248-250: ‘However, the changes were not observed in the 15N spin relaxation analysis, presumably because the time scale of the fluctuation is not sensitive to the 15N relaxation.’ – Should this be the other way around? i.e. the 15N relaxation is not sensitive to the time scale of the fluctuation?

Line 76 ‘intertwined dimer’ – does this situation qualify as a domain swap dimer? Would this be a better term to use?

Experimental design

The experimental work looks very sound and thorough, and all methods were well described.

Validity of the findings

The vast majority of their findings were robust and well stated.

I did identify a potential problem in the analysis of the predicted interaction with TBDTs, which influenced some of their conclusions. I would recommend they look at these conclusions again before publication. Specifically this relates to line 267 and Figure 6. Here the authors have shown the Paeruginosa TonB sequence (beta-5 strand) and how it might bind to the Ecoli BtuB sequence. They used the Ecoli sequence since there is no structure available for the Paeruginosa BtuB or other TBDT. However, there is a sequence (as opposed to structure) available for the Paeruginosa BtuB protein, and a sequence alignment could be carried out to look at the actual interacting residues in the correct species. I carried out my own such alignment, and I think some of the conclusions in this section may change when this sequence is used (for example, that in line 312, as well as others near line 267).

It would also be useful to state In the results (around line 188) what the similarity/identity of the PaeruginosaTonB sequence is to that of Ecoli.

Additional comments

There are a number of typos/grammatical errors that should be cleaned up prior to publication:

Line 32 ‘terminal terminus’ – either of these terms would be fine alone, but one needs to be removed.
Line 76 ‘composed’ – should be ‘were composed’.
Line 81 ‘as observed NMR’ should be ‘as observed in NMR’
Line 86 ‘the significant’ should be ‘a significant’
Line 88-89 ‘The major differences between the monomeric EcTonB without intertwining locate at residues 235-239 in the C-terminal end (Fig. 1b).’ – Can this sentence be clarified? E.g. ‘The major differences between the monomeric EcTonB and the intertwining dimer, pertain to residues 235-239 in the C-terminal end (Fig. 1b).’
Line 89-92 – ‘In the NMR structure of EcTonB-137, these residues fold to an additional beta-strand (beta6) and join in an anti-parallel fashion with the preceding beta-strand. On the other hand, the C-terminal end is either extended or largely invisible in the crystal structures (Fig. 1b). ‘ Would be better written as: In the NMR structure of EcTonB-137, these residues fold into an additional beta-strand (beta6) and join in an anti-parallel fashion with the preceding beta-strand, whereas in the crystal structures of monomeric EcTonB constructs, the C-terminal end is either extended or largely invisible (Fig. 1b).
Line 97 ‘On the other hand’ – would be better as ‘In constrast,’
Line 222 ‘by additional’ should be ‘by the additional’
Line 252 ‘affect to the effective’ should be ‘affect the effective’
Line 254 ‘with the lower order’ should be ‘with lower order’
Line 271 ‘phenylalanine of’ should be ‘phenylalanine in’
Line 285 ‘interfere the interaction site’ should be ‘interfere with the interaction site’
Line 316 ‘unveil structural’ should be ‘unveil the structural’

Reviewer 2 ·

Basic reporting

Overall the manuscript is well-written, but requires further proofreading to remove several minor errors.

Specific minor revisions needed:

The first paragraph of the Discussion introduces new modelling/docking information that would be more appropriate in the Results section.

The are number of citations to Ollila et al. 2018, which although submitted is still unpublished material. I'm unsure that this is acceptable according to PeerJ guidelines.

The HSQC depicted in Figure 2 requires darker contours.

L99 – A reference needed to specify exactly what crystal structures are referred to (dimers in crystal structures or complex of TonB Box and -terminus).

L179 – A reference for the Redfield equations would be useful.

L226/227 – Additional background information on the dynamics probed by parameters T1, T2, NOE etc. (i.e., timescales) would be helpful to guide non-NMR experts through interpretations.

Experimental design

No comments.

Validity of the findings

L108 to 111 – The authors state that the PaTonB-96 had an additional three residues because of cloning. Can they please state explicitly what these residues are and where they are located? For instance, residues located on the C-terminus have a dominating influence on the dynamic and structural findings which are of central importance to study.

L168 – Specific details are needed regarding the criteria used to assess the equilibration of MD trajectories.

Additional comments

No comments.

Reviewer 3 ·

Basic reporting

The authors report an NMR structure of a pseudomonas TonB protein C-terminal domain fragment.

The TonB protein is part of a mechanism for active transport across the outer membrane of Gram-negative bacteria, using energy derived from interactions in the cytoplasmic membrane below. In particular, the C-terminal domain of TonB interacts with outer membrane transporter proteins, via a conserved TonB box motif, presumably enabling structural changes in the transporter protein. The interaction is believed to involve the plug domain, which serves as a gate for the transporter protein channel.

The authors report a monomer structure with an extended and flexible C-terminal tail, in contract to some previous X-ray and NMR studies on TonB constructs from E. coli.

--- Possible corrections:

Abstract: "outer membrane" for "outermembrane"

89: are located

136: citations for VnmrJ and NMRPipe.

154: "chemical shift values ... were used ... for the structure calculation".

How? If additional software was used, citations are needed.

172: Lennard-Jones

195: were assigned

249: presumably because 15N relaxation is not sensitive to fluctuations at that time scale.

299: is located

Experimental design

The authors perform an NMR structure determination according to accepted methods, using NOE distances. They also use an appropriate set of NMR experiments to characterize backbone dynamics, and supplement these with molecular dynamics simulations.

Validity of the findings

The reported structure has good verification statistics, so the authors have made a good case that their structure is valid.

The molecular dynamics simulations correspond well with the NMR relaxation parameters, supporting the authors' reasonable claim that the protein has a flexible C-terminal end.

---

## Round 0.2 · accepted · Accept

All three reviewers are pleased with your response and revised manuscript.

# ·

Basic reporting

No comment

Experimental design

No comment

Validity of the findings

No comment

Additional comments

I am happy with the modifications made by the authors, and now recommend the paper for publication.

Reviewer 2 ·

Basic reporting

The authors have done a great job in addressing all of my previous suggestions. I have no further comments to add.

Experimental design

no comment

Validity of the findings

no comment.

Additional comments

no comment.

Reviewer 3 ·

Basic reporting

The authors report an NMR structure of a pseudomonas TonB protein C-terminal domain fragment. They have enhanced their report with a description of TonB-dependent energy transduction system.

The TonB protein is part of a mechanism for active transport across the outer membrane of Gram-negative bacteria, using energy derived from interactions in the cytoplasmic membrane below. In particular, the C-terminal domain of TonB interacts with outer membrane transporter proteins, via a conserved TonB box motif, presumably enabling structural changes in the transporter protein. The interaction is believed to involve the plug domain, which serves as a gate for the transporter protein channel.

The authors report a monomer structure with an extended and flexible C-terminal tail, in contract to some previous X-ray and NMR studies on TonB constructs from E. coli.

Experimental design

The authors perform an NMR structure determination according to accepted methods, using NOE distances. They also use an appropriate set of NMR experiments to characterize backbone dynamics, and supplement these with molecular dynamics simulations.

Validity of the findings

The reported structure has good verification statistics, so the authors have made a good case that their structure is valid.

The molecular dynamics simulations correspond well with the NMR relaxation parameters, supporting the authors' reasonable claim that the protein is monomeric and has has a flexible C-terminal end.